# Michaelis-Arbuzov-Type Reaction of 1-Imidoalkyltriarylphosphonium Salts with Selected Phosphorus Nucleophiles

**DOI:** 10.3390/molecules24183405

**Published:** 2019-09-19

**Authors:** Jakub Adamek, Anna Węgrzyk-Schlieter, Klaudia Steć, Krzysztof Walczak, Karol Erfurt

**Affiliations:** 1Department of Organic Chemistry, Bioorganic Chemistry and Biotechnology, Silesian University of Technology, B. Krzywoustego 4, 44-100 Gliwice, Polandstec-k@o2.pl (K.S.); krzysztof.walczak@polsl.pl (K.W.); 2Biotechnology Center of Silesian University of Technology, B. Krzywoustego 8, 44-100 Gliwice, Poland; 3Department of Chemical Organic Technology and Petrochemistry, Silesian University of Technology, B. Krzywoustego 4, 44-100 Gliwice, Poland; karol.erfurt@polsl.pl

**Keywords:** organophosphorus chemistry, phosphonium salts, phosphonates, phosphinates, phosphine oxides

## Abstract

In this study, Michaelis-Arbuzov-type reaction of 1-imidoalkyltriarylphosphonium salts with phosphites, phosphonites, and phosphinites was used in the synthesis of a wide range of phosphorus analogs of α-amino acids such as 1-imidoalkylphosphonates, 1-imidoalkylphosphinates, and 1-imidoalkylphosphine oxides. Large differences were observed in the reactivity of substrates depending on their structure, especially on the type of phosphonium moiety and *N*-protecting group. The conditions under which the expected products can be obtained in good to excellent yields have been developed. Mechanistic aspects of the transformation have been provided.

## 1. Introduction

The chemistry of phosphorus analogs of α-amino acids has enjoyed unflagging interest for decades [1,2,3,4,5,6,7,8,9,10]. The replacement of flat carboxylic moiety with either the phosphonyl, phosphinyl, or phosphinoyl group in the structure of α-amino acids enables them to have extremely valuable biological properties. These compounds are strong bioregulators and are used in agrochemicals as growth regulators, fungicides, and insecticides; they show antiviral, antibacterial, and anticancer properties [1,2,3,6,9,11,12,13].

1-Imidoalkylphosphonates, 1-imidoalkylphosphinates, and 1-imidoalkylphosphine oxides can be classified as a specific group of phosphorus analogs of α-amino acids. They are used as building blocks in the synthesis of many bioactive compounds such as phosphopeptides and statins (which act as enzyme inhibitors) [14,15], oligonucleotides [16], cytotoxic agents (e.g., cryptophycin 52) [17], and herbicides and plant growth regulators (2,4,5-imidazolidinetriones) [18]. Moreover, they are used in the synthesis of macrocyclic ligands, which are useful in medical diagnostics for obtaining luminescent probes and contrast agents in magnetic resonance imaging (MRI) [19,20]. This type of phosphorus analogs of α-amino acids exhibits also antibacterial and antifungal properties [21].

The synthesis of 1-imidoalkylphosphonates, 1-imidoalkylphosphinates, and 1-imidoalkylphosphine oxides described in the literature are mainly based on the Michaelis-Arbuzov- or Michaelis-Becker-type reaction of the corresponding halides with phosphorus nucleophiles, and in most cases, they are related to the simplest derivatives (**1**, R^1^ = H; I and II, Scheme 1) [15,19,22,23,24,25,26,27,28,29]. Other known methods are rather labor consuming and require more sophisticated substrates such as 1-hydroxyalkylphosphonates [30] or silyl compounds [31] (Scheme 1, I, path D and E), which limits the structural diversity of the products obtained. Scheme 1 presents a fused summary of the various approaches discussed above.

In 2017, we described a three-step protocol for the synthesis of 1-imidoalkyltriarylphosphonium salts from α-amino acids and demonstrated their reactivity as imidoalkylating agents on the example of reaction with aromatic compounds [32,33]. In this study, we present the results of our research on the synthetic utility of 1-imidoalkyltriarylphosphonium salts in the Michaelis-Arbuzov-type reaction with phosphorus nucleophiles (III, Scheme 1).

## 2. Results and Discussion

The reactivity of 1-imidoalkylphosphonium salts toward phosphorus nucleophiles was investigated based on the reaction of 1-(phthalimido)ethyltris(4-trifluoromethylphenyl)phosphonium salt **2a** with trimethyl phosphite (Scheme 2).

We assumed that in the first stage of the reaction, it is necessary to cleave the C_α_–P^+^ bond and form a corresponding carbenium cation **3a** (Scheme 2). Our previous research has shown that depending on the type of 1-imidoalkylphosphonium salt, the cations can be efficiently generated at a high temperature, which is appropriate for that of salt. The appropriate temperature for tris(4-trifluoromethylphenyl)phosphonium salts was about 90–100 °C [32]. Therefore, we decided to conduct the first experiments at 100 °C. Indeed, it was confirmed that the reaction proceeds at an acceptable rate at this temperature (reaction time was 2 h; the minimum temperature at which any visible progress of the reaction observed was equal to 80 °C).

Then, we investigated the effect of molar ratio of substrates and the type of solvent on the reaction yield (entries 1–5, Table 1). It should be noted that one of the Michaelis-Arbuzov reaction stages is dealkylation of the alkoxyphosphonium salt **4** (see Scheme 3) [34,35]. To facilitate the course of this step, we used methyltriphenylphosphonium iodide as the catalyst (iodide anions are a good dealkylating agent) [34,35,36]. The progress of the reaction was monitored based on ^1^H-NMR spectroscopy. The highest yields were obtained by conducting the reaction in chloroform at a molar ratio of reagents equal to 1:10:0.25 (salt **2**:P-nucleophile:catalyst). We also proved that the addition of the catalyst was not necessary, but it improved the yields (compare entries 6 and 7, Table 1). In turn, the change in the polarity of the solvent did not give any positive effects on the reaction (compare entries 3–5, Table 1).

Next, we determined the scope of applicability of the developed method. To this end, we modified the structure of 1-imidoalkylphosphonium salt (phosphonium moiety, *N*-protecting group, and substituent R^1^ at the α-position) and changed the type of phosphorus nucleophile.

Modifications in the phosphonium moiety (the introduction of electron-withdrawing groups) and changes of the *N*-protecting group had the greatest influence on the course of the examined reaction. The highest yields were obtained for 1-(*N*-phthalimido)alkylphosphonium salt derivatives of tris(3-chlorophenyl)phosphine and tris(4-trifluorophenyl)phosphine. Based on our results, 1-(*N*-succinimido)alkylphosphonium salts showed significantly less reactivity. To increase the yield, we had to raise the temperature and increase the concentration of phosphorus nucleophile (entry 18, Table 1). The reaction of triphenylphosphonium salt **2c** also required higher temperature, but in this case, the yield was still poor (entry 9, Table 1). 

Changes of substituents in the α-position did not significantly affect reaction conditions or yields. Only in the case of phosphonium salts **2d** and **2e** (no possibility of the side β-elimination reaction), higher yields of the expected products were noticed, which was particularly apparent in the preparation of 1-imidoalkylphosphinates and 1-imidoalkylphosphine oxides (entries 21 and 24, Table 1).

In the end, we proved that the described protocol could be used not only in reactions with phosphites but also with other phosphorus nucleophiles such as phosphonites and phosphinites (entries 19–24, Table 1). The expected products—1-imidoalkylphosphinates and 1-imidoalkylphosphine oxides—were obtained with satisfactory to excellent yield.

Based on our previous research [32,33,34,35,36] and above-described facts, we proposed a plausible mechanism of the Michaelis-Arbuzov-type reaction of 1-imidoalkylphosphonium salts with phosphorus nucleophiles (Scheme 3). It seems that the first stage is crucial for the course of the entire reaction. In the case of 1-imidoalkylphosphonium salts with weakened C_α_–P^+^ bond strength (Ar = *m*-C_6_H_4_Cl, *p*-C_6_H_4_CF_3_), the generation of imidocarbenium cation **3** was easier and the reaction temperature was lower (80–120 °C) than that of 1-imidoalkyltriphenylphosphonium salt (Ar = Ph, 150 °C). Moreover, the high temperature promoted side reactions—the formation of enimides **5** by β-elimination. All these factors affect the efficiency of the synthesis. Yields were much higher when 1-imidoalkylphosphonium salts with weakened C_α_–P^+^ bond strength were used as substrates (see Table 1).

In the second stage, the alkoxyphosphonium salt **4** was formed as a result of nucleophilic attack by phosphorus (PR^2^R^3^OR, Scheme 3). Dealkylation of this intermediate led to the expected products: 1-imidoalkylphosphonates, 1-imidoalkylphosphinates, and 1-imidoalkylphosphine oxides. The addition of catalyst facilitated this stage and improved the yield of the reaction. However, at the elevated temperature, phosphines generated in the first step could also act as the dealkylating agent. We think it explains the possibility of conducting the reaction without a catalyst (but with lower yields).

## 3. Experimental Section

### 3.1. General Information

Melting points were determined in capillaries and were uncorrected. Next, ^1^H- and ^13^C-NMR spectra were recorded at operating frequencies of 400 and 100 MHz, respectively, using tetramethylsilane (TMS) as the resonance shift standard. ^31^P-NMR spectra were recorded at operating frequencies of 161.9 MHz without the resonance shift standard, with respect to H_3_PO_4_ as 0 ppm. All chemical shifts (δ) are reported in ppm and coupling constants (*J*) in Hz. IR-spectra were measured on an FT-IR spectrophotometer (attenuated total reflectance method; ATR). High-resolution mass spectrometry (HR-MS) analyses were performed on a Waters Xevo G2 Q-TOF mass spectrometer equipped with an electrospray ionization (ESI) source operating in the positive ion mode. The accurate mass and composition of the molecular ion adducts were calculated using the MassLynx software incorporated within the instrument.

### 3.2. Synthesis

#### 3.2.1. Substrate Synthesis

The synthesis of 1-imidoalkyltriarylphosphonium salts **2** from α-amino acids was performed according to our previously described three-step procedure [32].

#### 3.2.2. General Procedure for the Reaction of 1-Imidoalkyltriarylphosphonium Salts **2** with Phosphorus Nucleophiles

To a suspension of 1-imidoalkyltriarylphosphonium salt **2** (0.25 mmol, 1.0 equiv.) in CHCl_3_ (2 cm^3^) placed in a glass vial sealed with a screw-cap a phosphorus nucleophile (2.5 mmol, 10 equiv.) and, in the case of a catalytic reaction, methyltriphenylphosphonium iodide (25.3 mg, 0.0625 mmol, 0.25 equiv.) was added. The reaction was carried out under conditions given in Table 1. Then, the solvent was evaporated to dryness under reduced pressure and the product was isolated by column chromatography using 50 cm^3^ of hexane and then ethyl acetate as the eluent. If necessary, purification can be repeated using acetonitrile or toluene:ethyl acetate (5:1, *v*/*v*) as the eluent (this is particularly useful during the separation of diastereoisomers **1h** and **1i**). The crystalline compounds were recrystallized from toluene.

*Dimethyl 1-(N-Phthalimido)ethylphosphonate* (**1a**) [37]. White crystals (53.8 mg, 76% yield), m.p. 75.0–77.0 °C. ^1^H-NMR (400 MHz, CD_3_CN) δ 7.71–7.80 (m, 4H, Phth), 4.59 (dq, *J*_1_ = 18.7 Hz, *J*_2_ = 7.6 Hz, 1H, C_α_H), 3.69 (d, *J* = 10.7 Hz, 3H, OCH_3_), 3.63 (d, *J* = 10.7 Hz, 3H, OCH_3_), 1.57 (dd, *J*_1_ = 16.3 Hz, *J*_2_ = 7.6 Hz, 3H, CH_3_) ppm; ^13^C-NMR (100 MHz, CD_3_CN) δ 168.4 (C=O), aromatic carbons: 135.6, 132.7, 124.2, 54.1 (d, *J* = 6.5 Hz, OCH_3_), 53.8 (d, *J* = 6.8 Hz, OCH_3_), 43.2 (d, *J* = 158.0 Hz, C_α_H), 13.5 (CH_3_) ppm; ^31^P-NMR (161.9 MHz, CD_3_CN) δ 25.1 ppm; IR (ATR) 2960, 1709, 1380, 1280, 1253, 1021, 906, 877, 832, 806, 712, 681 cm^−1^ (Appendix A).

*Dimethyl N-Phthalimidomethylphosphonate* (**1b**) [24]. White crystals (63.9 mg, 95% yield), m.p. 145.5–147.5 °C. ^1^H-NMR (400 MHz, CD_3_CN) δ 7.84–7.89 (m, 2H, Phth), 7.80–7.84 (m, 2H, Phth), 4.06 (d, *J* = 11.2 Hz, 2H, CH_2_), 3.75 (d, *J* = 10.9 Hz, 6H, 2×OCH_3_) ppm; ^13^C-NMR (100 MHz, CD_3_CN) δ 168.2 (C=O), aromatic carbons: 135.6, 132.8, 124.3, 54.0 (d, *J* = 6.1 Hz, OCH_3_), 33.2 (d, *J* = 157.6 Hz, CH_2_) ppm; ^31^P-NMR (161.9 MHz, CD_3_CN) δ 22.1 ppm; IR (ATR) 2974, 1715, 1403, 1238, 1060, 1014, 903, 845, 715, 701 cm^−1^.

*Diethyl 1-(N-Phthalimido)ethylphosphonate* (**1c**) [30]. Resin (70.0 mg, 90% yield). ^1^H-NMR (400 MHz, CD_3_CN) δ 7.75–7.88 (m, 4H, Phth), 4.61 (dq, *J*_1_ = 18.5 Hz, *J*_2_ = 7.6 Hz, 1H, C_α_H), 4.00–4.20 (m, 4H, 2×OCH_2_CH_3_), 1.66 (dd, *J*_1_ = 16.1 Hz, *J*_2_ = 7.6 Hz, 3H, CH_3_), 1.28 (td, *J*_1_ = 7.1 Hz, *J*_2_ = 0.5 Hz, 3H, OCH_2_CH_3_), 1.23 (td, *J*_1_ = 7.0 Hz, *J*_2_ = 0.5 Hz, 3H, OCH_2_CH_3_) ppm; ^13^C-NMR (100 MHz, CD_3_CN) δ 168.4 (d, *J* = 1.8 Hz, C=O), aromatic carbons: 135.5, 132.7, 124.1, 63.7 (d, *J* = 6.5 Hz, OCH_2_CH_3_), 63.4 (d, *J* = 6.5 Hz, OCH_2_CH_3_), 43.9 (d, *J* = 158.0 Hz, C_α_H), 16.8 (d, *J* = 5.7 Hz, OCH_2_CH_3_), 13.4 (CH_3_) ppm; ^31^P-NMR (161.9 MHz, CD_3_CN) δ 22.6 ppm; IR (ATR) 2989, 1716, 1383, 1010, 906, 879, 719 cm^−1^.

*Diethyl Phenyl(N-phthalimido)methylphosphonate* (**1d**) [38]. White crystals (92.4 mg, 99% yield), m.p. 102.5–104.5 °C. ^1^H-NMR (400 MHz, CD_3_CN) δ 7.77–7.90 (m, 4H, Ph), 7.56–7.66 (m, 2H, Ph), 7.28–7.41 (m, 3H, Ph), 5.70 (d, *J* = 24.6 Hz, 1H, C_α_H), 4.00–4.19 (m, 4H, 2×OCH_2_CH_3_), 1.21 (t, *J* = 7.1 Hz, 3H, OCH_2_CH_3_), 1.20 (t, *J* = 7.0 Hz, 3H, OCH_2_CH_3_) ppm; ^13^C-NMR (100 MHz, CD_3_CN) δ 168.4 (d, *J* = 3.3 Hz, C=O), aromatic carbons: 135.7, 134.7, 132.5, 130.3, 129.5, 129.3, 124.4, 64.3 (d, *J* = 6.6 Hz, OCH_2_CH_3_), 63.6 (d, *J* = 6.9 Hz, OCH_2_CH_3_), 52.3 (d, *J* = 156.9 Hz, C_α_H), 16.7 (d, *J* = 5.3 Hz, OCH_2_CH_3_), 16.6 (d, *J* = 5.3 Hz, OCH_2_CH_3_) ppm; ^31^P-NMR (161.9 MHz, CD_3_CN) δ 18.4 ppm; IR (ATR) 2988, 1712, 1383, 1359, 1254, 1053, 1026, 1014, 969, 890, 802, 728, 719, 697 cm^−1^.

*Diethyl 3-Methyl-1-(N-phthalimido)butylphosphonate* (**1e**). Resin (83.0 mg, 94% yield). ^1^H-NMR (400 MHz, CD_3_CN) δ 7.80–7.87 (m, 4H, Phth), 4.58 (ddd, *J*_1_ = 19.6 Hz, *J*_2_ = 11.9 Hz, *J*_3_ = 4.3 Hz, 1H, C_α_H), 4.03–4.17 (m, 4H, 2xOCH_2_CH_3_), 2.39–2.49 (m, 1H, CH), 1.60–1.69 (m, 1H, CH), 1.48–1.55 (m, 1H, CH), 1.27 (td, *J*_1_ = 7.1 Hz, *J*_2_ = 0.5 Hz, 3H, OCH_2_CH_3_), 1.23 (td, *J*_1_ = 7.0 Hz, *J*_2_ = 0.5 Hz, 3H, OCH_2_CH_3_), 0.91 (d, *J* = 6.7 Hz, 3H, CH_3_), 0.88 (d, *J* = 6.5 Hz, 3H, CH_3_) ppm; ^13^C-NMR (100 MHz, CD_3_CN) δ 168.7 (d, *J* = 1.4 Hz, C=O), aromatic carbons: 135.6, 132.5, 124.2, 63.6 (d, *J* = 6.5 Hz, OCH_2_CH_3_), 63.3 (d, *J* = 6.6 Hz, OCH_2_CH_3_), 46.8 (d, *J* = 156.2 Hz, C_α_H), 35.9 (d, *J* = 1.2 Hz, CH_2_), 25.7 (d, *J* = 12.3 Hz, CH), 23.2 (OCH_2_CH_3_), 21.2 (OCH_2_CH_3_), 16.8 (CH_3_), 16.7 (CH_3_) ppm; ^31^P-NMR (161.9 MHz, CD_3_CN) δ 22.3 ppm; IR (ATR) 2959, 1713, 1379, 1249, 1051, 1017, 963, 717 cm^−1^; HRMS (ESI-TOF) calculated for C_17_H_25_NO_5_P [M + H]^+^ 354.1470 found 354.1469.

*Diethyl 1-(N-Succinimido)ethylphosphonate* (**1g**). Resin (42.8 mg, 65% yield). ^1^H-NMR (400 MHz, CD_3_CN) δ 4.57 (dq, *J*_1_ = 19.1 Hz, *J*_2_ = 7.5 Hz, 1H, C_α_H), 4.04–4.30 (m, 4H, 2xOCH_2_CH_3_), 2.73 (s, 4H, 2xCH_2_), 1.57 (dd, *J*_1_ = 16.0 Hz, *J*_2_ = 7.5 Hz, 3H, CH_3_), 1.35 (t, *J* = 6.6 Hz, 3H, OCH_2_CH_3_), 1.33 (t, *J* = 6.8 Hz, 3H, OCH_2_CH_3_) ppm; ^13^C-NMR (100 MHz, CD_3_CN) δ 176.4 (C=O), 63.4 (d, *J* = 6.4 Hz, OCH_2_CH_3_), 62.4 (d, *J* = 6.6 Hz, OCH_2_CH_3_), 43.6 (d, *J* = 158.2 Hz, C_α_H), 28.2 (CH_2_), 16.5 (OCH_2_CH_3_), 16.5 (OCH_2_CH_3_), 13.0 (CH_3_) ppm; ^31^P-NMR (161.9 MHz, CD_3_CN) δ 22.4 ppm; IR (ATR) 2984, 1702, 1387, 1238, 1195, 1051, 1017, 960, 797 cm^−1^; HRMS (ESI-TOF) calculated for C_10_H_19_NO_5_P [M + H]^+^ 264.1001 found 264.1001.

*Methyl phenyl [1-(N-Phthalimido)ethyl]phosphinate* (**1ha + 1hb**). A mixture of two diastereoisomers, resin (48.6 mg, 59% yield). ^1^H-NMR (400 MHz, CD_3_CN) δ 7.78–7.81 (m, 2H, Ph)^a^, 7.58–7.77 (m, 4H, Ph)^a^, 7.37–7.56 (m, 3H, Ph)^a^, 4.68–4.80 (m, 1H, C_α_H)^a^, 3.68 (d, *J* = 10.8 Hz, 3H, OCH_3_)^b^, 3.66 (d, *J* = 10.7 Hz, 3H, OCH_3_)^b^, 1.69 (dd, *J*_1_ = 14.5 Hz, *J*_2_ = 7.6 Hz, 1H, CH_3_)^b^, 1.60 (dd, *J*_1_ = 15.5 Hz, *J*_2_ = 7.6 Hz, 2H, CH_3_)^b^ ppm; ^13^C-NMR (100 MHz, CD_3_CN) δ 168.4 (d, *J* = 1.2 Hz, C=O), 168.2 (d, *J* = 1.1 Hz, C=O), aromatic carbons: 135.5, 135.4, 133.8 (d, *J* = 2.7 Hz), 133.8 (d, *J* = 2.8 Hz), 133.6 (d, *J* = 9.9 Hz), 133.1 (d, *J* = 9.9 Hz), 132.7, 132.5, 130.2 (d, *J* = 18.3 Hz), 129.6, 129.5, 129.0 (d, *J* = 18.3 Hz), 124.0, 123.9, 52.5 (d, *J* = 6.5 Hz, OCH_3_), 52.4 (d, *J* = 6.1 Hz, OCH_3_), 46.8 (d, *J* = 109.0 Hz, C_α_H), 46.7 (d, *J* = 108.0 Hz, C_α_H), 13.1 (d, *J* = 2.7 Hz, CH_3_), 12.6 (d, *J* = 1.3 Hz, CH_3_) ppm; ^31^P-NMR (161.9 MHz, CD_3_CN) δ 39.3, 39.0 ppm; IR (ATR) 2947, 1711, 1378, 1226, 1120, 1023, 992, 879, 791, 718, 697 cm^−1^; HRMS (ESI-TOF) calculated for C_17_H_17_NO_4_P [M + H]^+^ 330.0895 found 330.0898. ^a^Overlapping signals of both diastereoisomers. ^b^Separate signals of both diastereoisomers.

*Methyl Phenyl[phenyl(N-phthalimido)methyl]phosphinate* (**1i**). A mixture of two diastereoisomers (92.0 mg, 94% yield). Diastereoisomers were separated by column chromatography (toluene:ethyl acetate; 5:1, *v*/*v*). 

*Methyl Phenyl[phenyl(N-phthalimido)methyl]phosphinate* (**1ia**, **the first diastereoisomer**). Resin. ^1^H-NMR (400 MHz, CD_3_CN) δ 7.71–7.75 (m, 4H, Ph), 7.63–7.71 (m, 5H, Ph), 7.44–7.52 (m, 1H, Ph), 7.33–7.43 (m, 4H, Ph), 5.83 (d, *J* = 17.9 Hz, 1H, C_α_H), 3.63 (d, *J* = 11.0 Hz, 3H, OCH_3_) ppm; ^13^C-NMR (100 MHz, CD_3_CN) δ 168.1 (d, *J* = 2.6 Hz, C=O), aromatic carbons: 135.6, 134.2, 133.8 (d, *J* = 2.9 Hz), 133.1 (d, *J* = 9.9 Hz), 132.2, 131.1, 131.0, 129.5 (d, *J* = 12.7 Hz), 129.5 (d, *J* = 0.8 Hz), 129.5, 124.2, 55.1 (d, *J* = 107.3 Hz, C_α_H), 52.6 (d, *J* = 6.5 Hz, OCH_3_) ppm; ^31^P-NMR (161.9 MHz, CD_3_CN) δ 36.3 ppm; IR (ATR) 3059, 1717, 1379, 1070, 1033, 717, 696 cm^−1^; HRMS (ESI-TOF) calculated for C_22_H_19_NO_4_P [M + H]^+^ 392.1052 found 392.1054. 

*Methyl Phenyl[phenyl(N-phthalimido)methyl]phosphinate* (**1ib**, **the second diastereoisomer**). Resin. ^1^H-NMR (400 MHz, CD_3_CN) δ 7.80–7.90 (m, 4H, Ph), 7.42–7.55 (m, 5H, Ph), 7.31–7.40 (m, 2H, Ph), 7.23–7.30 (m, 3H, Ph), 5.87 (d, *J* = 21.5 Hz, 1H, C_α_H), 3.63 (d, *J* = 11.0 Hz, 3H, OCH_3_) ppm; ^13^C-NMR (100 MHz, CD_3_CN) δ 168.7 (d, *J* = 2.5 Hz, C=O), aromatic carbons: 135.7, 134.5 (d, *J* = 2.5 Hz), 134.0 (d, *J* = 9.1 Hz), 133.6 (d, *J* = 2.8 Hz), 132.6, 130.3, 130.2, 129.5 (d, *J* = 1.6 Hz), 129.3 (d, *J* = 2.2 Hz), 129.0 (d, *J* = 13.0 Hz), 124.4, 55.6 (d, *J* = 104.5 Hz, C_α_H), 52.5 (d, *J* = 6.7 Hz, OCH_3_) ppm; ^31^P-NMR (161.9 MHz, CD_3_CN) δ 36.7 ppm; IR (ATR) 3060, 1716, 1381, 1236, 1071, 1024, 889, 791, 717, 695 cm^−1^; HRMS (ESI-TOF) calculated for C_22_H_19_NO_4_P [M + H]^+^ 392.1052 found 392.1052.

*Diphenyl 1-(N-Phthalimido)ethylphosphine oxide* (**1j**). White crystals (44.1 mg, 47% yield), m.p. 138.0–140.0 °C. ^1^H-NMR (400 MHz, CD_3_CN) δ 7.89–7.99 (m, 2H, Ph), 7.67–7.76 (m, 6H, Ph), 7.54–7.66 (m, 3H, Ph), 7.29–7.44 (m, 3H, Ph), 5.27 (qd, *J*_1_ = 7.5 Hz, *J*_2_ = 6.5 Hz, 1H, C_α_H), 1.69 (dd, *J*_1_ = 14.0 Hz, *J*_2_ = 7.5 Hz, 3H, CH_3_) ppm; ^13^C-NMR (100 MHz, CD_3_CN) δ 168.4 (d, *J* = 1.0 Hz, C=O), aromatic carbons: 135.4, 133.3 (d, *J* = 2.7 Hz), 133.0 (d, *J* = 2.8 Hz), 132.9 (d, *J* = 18.6 Hz), 132.4, 132.4 (d, *J* = 8.8 Hz), 131.9 (d, *J* = 21.6 Hz), 131.8 (d, *J* = 9.2 Hz), 129.9 (d, *J* = 11.4 Hz), 129.4 (d, *J* = 11.6 Hz), 123.9, 47.2 (d, *J* = 74.7 Hz, C_α_H), 12.9 (d, *J* = 1.9 Hz, CH_3_) ppm; ^31^P-NMR (161.9 MHz, CD_3_CN) δ 28.7 ppm; IR (ATR) 3057, 1710, 1376, 1202, 1116, 1046, 878, 715, 698, 669 cm^−1^; HRMS (ESI-TOF) calculated for C_22_H_19_NO_3_P [M + H]^+^ 376.1103 found 376.1104.

*Diphenyl Phenyl(N-phthalimido)methylphosphine oxide* (**1k**). White crystals (96.2 mg, 88% yield), m.p. 233.0–235.0 °C. ^1^H-NMR (400 MHz, CD_3_CN) δ 7.82–7.90 (m, 2H, Ph), 7.71–7.79 (m, 2H, Ph), 7.56–7.68 (m, 6H, Ph), 7.32–7.50 (m, 6H, Ph), 7.21–7.30 (m, 3H, Ph), 6.24 (d, *J* = 12.3 Hz, 1H, C_α_H) ppm; ^13^C-NMR (100 MHz, CD_3_CN) δ 167.5 (d, *J* = 2.2 Hz, C=O), aromatic carbons: 134.2, 133.0, 132.2 (d, *J* = 2.8 Hz), 132.0 (d, *J* = 3.2 Hz), 132.0 (d, *J* = 8.9 Hz), 131.8 (d, *J* = 7.9 Hz), 131.6 (d, *J* = 9.3 Hz), 130.9, 130.8, 130.8 (d, *J* = 9.5 Hz), 128.8 (d, *J* = 1.7 Hz), 128.8, 128.7 (d, *J* = 11.8 Hz), 128.4 (d, *J* = 11.9 Hz), 123.6, 55.4 (d, *J* = 71.2 Hz, C_α_H) ppm; ^31^P-NMR (161.9 MHz, CD_3_CN) δ 22.2 ppm; IR (ATR) 3058, 1717, 1372, 1345, 1321, 1199, 1119, 1101, 1074, 887, 709, 693 cm^−1^; HRMS (ESI-TOF) calculated for C_27_H_21_NO_3_P [M + H]^+^ 438.1259 found 438.1262.

## 4. Conclusions

The possibility of the synthetic use of 1-imidoalkyltriarylphosphonium salts in reactions with phosphorus nucleophiles was investigated. It has been noticed that the structure of the substrates has a great impact on the course of the studied reaction. Moreover, 1-(*N*-phthalimido)alkylphosphonium salts with weakened C_α_–P^+^ bond strength show the highest reactivity and react under milder conditions in comparison with other 1-imidoalkylpohosphonium salts. The reaction may be conducted with or without the addition of a catalyst. It has been demonstrated, that not only phosphites but also phosphonites, and phosphinites can be used as phosphorus nucleophiles. The most favorable reaction conditions for all examined types of 1-imidoalkylphosphonium salts and P-nucleophiles were proposed. A plausible reaction mechanism, which explains the currently obtained results, was also presented.

Our research confirmed a universal character of the developed method and its considerable potential in the synthesis of phosphorus analogs of α-amino acids.

Further studies on expanding the range of nucleophiles (not only P-nucleophiles), which can be used in a reaction with 1-imidoalkyltriarylphosphonium salts, are in progress.

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
