# Peer review of "Michaelis-Arbuzov-Type Reaction of 1-Imidoalkyltriarylphosphonium Salts with Selected Phosphorus Nucleophiles"

_molecules, 2019, doi:10.3390/molecules24183405_

Round 1
Reviewer 1 Report
This paper can be accept for its publication, but the Scheme 1 should be modified for a better understanding
Author Response
Reviewer #1:
First of all, thank you for your comments and suggestions that allowed us to greatly improve the quality of the manuscript.
1) This paper can be accept for its publication, but the Scheme 1 should be modified for a better understanding
Response: Scheme 1 has been changed according to the reviewer’s comment.
Reviewer 2 Report
The ms of Adamek et al is on a novel Arbuzov-like reaction of imidoalkyltriarylphosphonium salts with >P(OR) derivatives.
The ms will be suitable for publication in Molecules after the following minor revision - Abstract: pls remove the trivial spectral identification (last) sentence. -Scheme 4: it is a good summary, but some parts are too small. Pls enlarge the parts written in small letters. Scheme2: Pls insert the by-products under the too arrows (“- P(Ar)3” and “- MeBF4”) Head of Table1: on the arrow “PR2R3R4” is awkward. It would be better to show at least one RO, eg “>POR” .(PR2R3OR” is much better in Scheme3). This Referee is not sure if the arrow starting from I- and directed to OR is OK. Instead, the arrow should start from BF4- and directed to OR. Assignment of the 13C NMR data would mean an added value. Pls assign!- What about adding MS values (instead of IR)?
Author Response
Reviewer 2
Thank you for your careful reading of the manuscript and helpful comments and suggestions. We have made revisions according to your comments and suggestions, as described below.
1) Abstract: pls remove the trivial spectral identification (last) sentence.
Response: It has been done according to the reviewer’s comment.
2) Scheme 4: it is a good summary, but some parts are too small. Pls enlarge the parts written in small letters.
Response: It has been done according to the reviewer’s comment.
3) Scheme2: Pls insert the by-products under the too arrows (“- P(Ar)3” and “- MeBF4”)
Response: It has been done according to the reviewer’s comment.
4) Head of Table1: on the arrow “PR2R3R4” is awkward. It would be better to show at least one RO, eg “>POR” .(PR2R3OR” is much better in Scheme3).
Response: It has been done according to the reviewer’s comment.
5) This Referee is not sure if the arrow starting from I- and directed to OR is OK. Instead, the arrow should start from BF4- and directed to OR.
Response:
The arrow shows the course of dealkylation - this is a nucleophilic substitution. In the examined reaction, the nucleophilic agent is either I- or phosphine (not BF4-!) – the arrow should start from I- and lead to OR. It should be noted that BF4- is a non-nucleophilic anion that cannot participate in this reaction.
7) Assignment of the 13C NMR data would mean an added value. Pls assign!
Response:
The assignment of the 1H NMR and 13C NMR has been added according to the reviewer’s comment.
8) What about adding MS values (instead of IR)?
Response:
MS values are given in spectroscopic characteristics (manuscript pages: 6-8). MS spectra have been added to the supporting materials
Best regards,
Jakub Adamek